# Occupational Health and Safety, Organisational Commitment, and Turnover Intention in the Spanish IT Consultancy Sector

**DOI:** 10.3390/ijerph18115658

**Published:** 2021-05-25

**Authors:** Julio Suárez-Albanchez, Juan Jose Blazquez-Resino, Santiago Gutierrez-Broncano, Pedro Jimenez-Estevez

**Affiliations:** 1Faculty of Law and Social Sciences, Castilla-La Mancha University, San Pedro Martir, 45071 Toledo, Spain; 2Department of Business Administration, Faculty of Social Science, Castilla-La Mancha University, Av. Real Fábrica de Seda, Talavera de la Reina, 45600 Toledo, Spain; Juan.Blazquez@uclm.es (J.J.B.-R.); Santiago.Gutierrez@uclm.es (S.G.-B.); 3Department of Business Administration, Faculty of Law and Social Science, Castilla-La Mancha University, San Pedro Martir, 45071 Toledo, Spain; Pedro.Jestevez@uclm.es

**Keywords:** health and safety, turnover intention, organisational commitment, work engagement, Covid-19 pandemic

## Abstract

Background: The purpose of this study is to analyse the impact that occupational health and safety policies have on employees’ work and organisational commitment and, in turn, on their intention to leave the company. Methods: For this study, we designed a questionnaire with a five-level Likert scale and distributed it among professionals from different companies in the IT consultancy sector in Spain. The data collected from 458 completed questionnaires were analysed using the partial least squares structural equation modelling (PLS-SEM) technique using the SmartPLS software. Results: From the analysis of the data, it was concluded that there is indeed a positive relationship between occupational health and safety policies and employees’ work and organisational commitment, as well as a negative relationship between these policies and the intention to leave the company. Similarly, there is a negative relationship between employees’ work and organisational commitment and their turnover intention. Conclusions: Although, due to the activity of professionals in the sector, occupational health and safety has not been an area of priority, it was concluded from this study that an improvement in these areas would have a beneficial effect on the commitment of workers to the company, thus helping to reduce the high levels of turnover in the sector. Future lines of research, as well as their practical application and the limitations of the study, are indicated at the end of the paper.

## 1. Introduction

The long-term success of any organisation depends on the retention of key employees. To a large extent, factors such as customer satisfaction, organisational performance, and a good work climate depend on the organisation’s ability to retain the best employees [1]. A report by the Society for Human Resource Management SHRM [2] states that 75% of employees are actively looking for another job; these results should alert employers to this potential problem.

Several aspects influence the well-being of workers and impact their performance, their degree of commitment to the company, and, above all, their intention to leave the company. In Spain, the latter had an average incidence of 21% in 2018, rising to 25.6% in the area of information and communication technologies [3]. Being aware of these aspects and understanding how they influence workers is necessary if we want to keep our staff motivated and retain the best talent. This is important in any company, but is even more so in a sector as dynamic and competitive as the IT consultancy sector, which has high turnover rates and requires high-level qualifications and professional experience that are often difficult to obtain.

According to the National Institute of Statistics’ [4] 2018 labour force survey, the information and communications technology sector had the highest employment rate (79.88%), making the retention of the best talent a vitally needed tool for this sector.

This paper analyses the impact that occupational health and safety has on organisational commitment and intention to leave the organisation on the part of professionals in the IT consultancy sector in Spain.

Although some of the risks associated with working with computer equipment are physical in nature [5], such as visual fatigue, physical fatigue, and other problems [6], studies have found that one of the main health problems among software developers is stress, which, in addition to being very common among these professionals, is highly problematic [7,8]. Stress has been found to be an important causal agent in health problems [9,10]. According to stress theory [11], stress diminishes performance as a result of its impact on the personal appraisal process, so it is important to reduce its incidence among professionals in the IT consulting area in order to improve their performance and maintain their health.

It is important to consider that the circumstances experienced in recent months as a result of the Covid-19 pandemic have led to an increase in health-related problems in workers, particularly regarding their mental health and experiences of stress [12,13,14]. This makes this study especially relevant given that it analyses a sector already affected by stress.

Despite the importance that healthcare, particularly in the area of mental health, has in the IT consulting sector in Spain, its impact on employee turnover has hardly been discussed.

### 1.1. Talent Retention in IT consultancy

According to data from the Spanish Association of Consultancy Firms [15], in 2019 this sector obtained revenues of 14,517 million euros, an increase of 5.9% over the previous year. Regarding employment, the sector saw an 8.9% increase in its workforce in 2019 compared with the previous year, exceeding 202,000 workers. It is important to highlight that more than 66% of workers in the sector have university-level qualifications, with 74% of graduates specialising in Science, Technology, Engineering, and Mathematics (STEM) areas. This is a sector of great importance to the national economy and employment, which is why it is important to understand which aspects are capable of motivating workers to commit to their companies and helping to reduce the high staff turnover rate, which currently stands at around 15% [16].

Currently, the problem of high levels of employee turnover is one of the main concerns among company managers due to the high time and money costs that staff training represents for organisations [17]. Therefore, ensuring that the most experienced and useful staff remain with the organisation must be a priority for any organisation.

This study focuses on analysing how occupational health and safety policies influence workers’ commitment to the organisation and their intention to leave the company. The Covid-19 pandemic has had a major impact on workers’ and companies’ concerns regarding health and safety in the workplace in sectors where, historically, concerns regarding this aspect have been secondary [18,19]. Although studies [20,21] have analysed the positive effects of promoting employee health and safety, there has been little research on the impact this aspect has on the organisational commitment and turnover intention of workers. Moreover, these studies mainly focused on high-risk professional sectors, so a novel line of research is to relate these variables in a low-hazard sector such as IT consultancy, especially if we consider the impact of Covid-19 on health.

This study aims to fill the gap in the literature and increase our current knowledge on how health and safety at work are related to the commitment to the company and the turnover intention of workers in the IT consultancy sector in Spain. The study analyses how companies should approach these aspects in order to reduce the high turnover in the sector.

To the aforementioned problem of high levels of turnover in the sector is added the problem caused by the Covid-19 pandemic, which has forced people to pay even more attention to health and safety at work. It is very important to analyse the impact that these policies have on the organisational commitment and turnover intention of workers in the sector and to understand how improvements to these policies can reduce this problem.

### 1.2. Job Safety and Employees’ Turnover Intention

In recent years, occupational health and safety has attracted the attention of companies worldwide as it is related to job satisfaction, worker productivity, organisational commitment, and employees’ turnover intention [20,22].

Staff turnover is the process by which an employee leaves his or her organisation [23]. This turnover can be either involuntary, when it is the company that decides to dispense with the employee’s services, or voluntary, when the employee leaves the organisation on their own initiative. In this study, it is referred to as voluntary turnover or the employee’s turnover intention, which has a negative impact on business efficiency and productivity and causes significant costs for organisations [24,25].

Although numerous studies have analysed the relationship between occupational health and safety policies and employees’ turnover intention [20,21,26], these studies focused on activities of a dangerous nature that have nothing to do with the professional activities of IT consultancy. However, it was considered that, despite being a low-risk activity, the implementation of adequate occupational health and safety policies has a negative impact on employees’ turnover intention.

**Hypothesis** **1** **(H1).***There is a relationship between occupational safety and health policies and employees’ turnover intention*.

### 1.3. Organisational Commitment Mediates the Relationship between Occupational Safety and Health Policies and Employees’ Turnover Intention

Organisational commitment has been referred to in the literature as the level of identification that an employee has with his or her organisation and the degree to which an employee is willing to remain with his or her company [27]. Other authors have referred to organisational commitment as the force that binds people to organisations [28]. Wright and Bonett [29] found that organisational commitment not only has a positive effect on the professional performance of employees, but that committed employees are willing to make additional efforts and perform other tasks such as helping their colleagues. The impact of organisational commitment on these behaviours outside of their duties may even be greater than the impact on their own performance. We can therefore affirm that working to increase and maintain organisational commitment is of vital importance to organisations, especially if it is considered that organisational commitment, together with job satisfaction, plays a very important role in employees’ intention to leave the company [30].

Although organisational commitment is a fairly commonly studied topic in academic research [31,32], little has been said about it in the context of IT consultancy.

Organisational commitment is a motivational construct [33] that involves a positive psychological state related to work and is a determinant in predicting employees’ job performance [34]. Therefore, it is extremely important that organisations are able to obtain this commitment from their employees, especially if it is considered that organisational commitment has the potential to develop and last in the long term [35] and also impacts upon the improvement of high-performance work practices and job performance [36].

In this study, we determined whether organisational commitment mediates the relationship between occupational health and safety and turnover intention. Most research focuses on the relationship between two variables, X and Y; mediation represents the inclusion of a third variable, Z, which modifies the relationship from X -> Y to X -> Z -> Y and modifies the significance of the initial relationship [37]. The difference between a moderating and a mediating variable has been thoroughly investigated [38,39]; a mediating variable is one that is in a causal sequence between two variables. In our work, we determined how organisational commitment has a partially mediating relationship between occupational health and safety and intention to leave the company.

Previous work has validated the existence of this mediation in the energy industry sector in Ghana [40]. In our hypothesis, we considered that the commitment of workers to their organisation acts as a mediator between the company’s occupational health and safety policies and the employees’ turnover intention in the low-hazard sector of IT consultancy in Spain.

**Hypothesis** **2** **(H2).***There is a mediating relationship of work and company commitment between occupational safety and health policies and employees’ turnover intention*.

There is also an important relationship between the health and safety policies of the company and the organisational commitment of workers to the company, a relationship previously studied in different professional sectors [21,26,41,42]. In this study, it is contended that the implementation of appropriate occupational health and safety policies will have a positive impact on employees’ work and emotional commitment to their organisation, even in low-risk activities such as IT consultancy [21,26,41,42].

**Hypothesis** **3** **(H3).***There is a relationship between health and safety policies and employees’ work and organisational commitment*.

Some studies have identified organisational commitment as one of the main factors in employee turnover itself [21,41].

Organisational commitment is, together with job satisfaction, a key predictor of employee turnover [43], so it is important to try to understand which factors are able to negatively influence these aspects.

Although this idea has been previously explored in different articles [44,45], these focus on other professional areas. Our study is focused on analysing the impact that an employee’s commitment to his or her company has on his or her intention to leave the organisation in the IT consultancy sector.

**Hypothesis** **4** **(H4).***There is a relationship between employees’ work and organisational commitment and their turnover intention*.

## 2. Materials and Methods

To carry out this study, we used a questionnaire consisting of 33 pre-questions with a five-level Likert scale ranging from ‘strongly disagree’ to ‘strongly agree’. The questionnaire was distributed among professionals employed at the main Spanish companies in the IT consultancy sector during November 2020. The questions used in this study have been validated in previous studies [46,47,48,49,50] and were adapted to the particularities of our study. Of the 33 questions in the questionnaire, five correspond to socio-demographic characteristics of the respondents. The other 28 questions were grouped into five constructs: job safety, management safety practices, safety programme [46], organisational commitment/work engagement [47,50], and turnover intention [48], which were used for the statistical analysis.

The survey form was distributed to 1000 professionals in the IT consultancy sector in Spain, and we received 458 responses. Once the collected questionnaires were analysed and refined, the final sample reached a total of 201 participants.

The research questionnaire is available in the Table A1, Appendix A.

The first stage consisted of analysing the different socio-demographic characteristics of the study participants. The results of the analysis can be found in the following section.

To carry out the evaluation of our hypotheses, a study was carried out using a model of structural equations. The model was empirically analysed using partial least squares structural equation modelling (PLS-SEM).

The study variables were grouped into five different constructs: job safety, management safety practices, safety programme, organisational commitment/work engagement, and turnover intention, which were used for the statistical analysis.

A quantitative model was used for the study and validated using the partial least squares structural equation modelling (PLS-SEM) method in the SmartPLS 3 software program [51] as well as IBM’s SPSS tool.

An analysis of a measurement model requires the study of the validity and reliability of the latent variables through their relationship with their associated measures. This analysis was performed through an assessment of individual and construct variables’ reliability, convergent validity, and discriminant validity.

Once the reliability and validity of the measured model was established, the structural model was evaluated through an analysis of the relationships between the different constructs defined in the study hypotheses.

The following sections of this work show the results obtained through the evaluation of the structural model.

Structural equation modelling (SEM), including PLS-SEM, is a second-generation multivariate data analysis technique capable of providing a higher level of confidence to investigations due to its statistical efficiency, which is achieved largely through the use of specialised software such as SmartPLS, VisualPLS, or the various packages available for R. This software allows for the simultaneous examination of the relationships between several dependent and independent variables and is widely used in social science research among other fields [52].

There has been a significant increase in the use of SEM in recent years. It is being increasingly used in a wide variety of disciplines, among which is strategic management [53].

Statistically, SEM methods represent an evolution of linear modelling procedures and are used to evaluate whether the analysed model is consistent with the data collected for validating the theory [54]. Although the most widely used approach in SEM is covariance-based SEM (CB-SEM), more and more researchers are using PLS-SEM to analyse structural equation models [55], and we use this method in the current research. SEM is a multivariate analytical method used to estimate complex relationships between several variables, even when these relationships are not directly observable [56].

The PLS-SEM technique is valid for use in both confirmatory and exploratory investigations [57].

## 3. Results

This model was empirically analysed using the PLS-SEM technique, as it is the most suitable technique for this type of study due to its predictive capability [58,59]. This method is currently considered to be the most developed of the variance-based systems for SEM and is applied in a wide range of disciplines. The analysis was carried out according to a two-step approach: evaluation of the measurement model and evaluation of the structural model.

Regarding the socio-demographic characteristics of the study participants, it was found that the majority (35.48%) of the professionals surveyed were between 30 and 39 years old, with the second-largest group (29.57%) being between 40 and 49 years old. Moreover, 19.35% of the professionals were between 18 and 29 years old, while 15.6% of the professionals were between 50 and 59 years old. In terms of gender, there were more men (64.4%) than women (35.6%) in the sample. In terms of the level of education, 75.12% of the professionals had a university-level education, 34.33% had a postgraduate-level education, 23.38% had a secondary-level education, and only 1.5% had a primary-level education. The vast majority (86.57%) of those surveyed had a permanent contract, compared with 13.43% who had a temporary contract. In terms of the length of service at the current company, the majority (53.73%) of those surveyed had been at their current company for less than 5 years, 21.89% had been in their current job for between 5 and 10 years, 10.94% had been in their current job for between 11 and 15 years, and only 13.44% of those surveyed had been in their current job for more than 15 years, which supports our premise of the high turnover in the sector.

### 3.1. Measurement Model

We used the PLS-SEM method with the SmartPLS software due to the complexity of the model and the reliability of this method and its predictive power [60,61].

The evaluation of the model was carried out by validating different model factors. The significance level of the model is 95%. The internal consistency of the model was assessed through Cronbach’s alpha and composite reliability, while the convergent validity of the model was assessed through the indicator reliability and the average variance extracted (AVE). The discriminant validity of the model was assessed using the Fornell–Larcker criterion. Cross-loadings between indicators and latent variables were also assessed. Finally, to validate the internal consistency of the model, we checked that all variables reached an adequate Cronbach’s alpha value [62], as can be seen in Table 1.

We also verified that the composite reliability (CR) values are adequate, since they reached values ranging from 0.862 to 0.960 [63], as can be seen in Table 1.

Convergent validity indicates that a set of indicators represents a single construct [64] and is validated by the average variance extracted (AVE), which must be equal to or greater than 0.50. As Table 1 shows, our model meets this requirement.

Discriminant validity indicates the extent to which a construct differs from others and is measured mainly by the Fornell–Larcker criterion. This criterion considers the amount of variance that a construct captures from its indicators (AVE), which must be greater than the variance that the construct shares with other constructs. Table 2 shows the observed values for our model.

The value of a cross-loading must be higher for its own variable than with the other variables evaluated in the model [65]. Table 2 shows that our model satisfies this criterion.

On the other hand, Table 1 also shows the loading factors of the different variables together with their mean values and standard deviation. The loading factors of all the variables have acceptable values, which confirms the validity of our model.

### 3.2. Structural Model

The SEM used for this research is shown in Figure 1. This model is used to simultaneously assess the relationships between the different constructs of the model.

For the proposed model, a second-order construct was created using the hierarchical components method.

Based on this analysis, it was observed that occupational health and safety has a positive influence on organisational and work commitment (0.500) as well as a negative impact on turnover intention (−0.207). It was also observed that organisational and work commitment has a significant negative relationship with turnover intention (−0.466).

### 3.3. Mediation Analysis

Mediation exists when a variable, the mediating variable, enhances or reduces the influence of a preceding variable on a dependent variable, thus modifying the magnitude of the relationship between the two variables [66]. In our case, the mediating variable is Organisational Commitment/Work Engagement, the preceding variable is Occupational Health and Safety, and the dependent variable is Turnover Intention.

Mediation can be considered partial when the inclusion of the mediating construct reduces the strength of the relationship between the independent variable and the dependent variable, but the relationship is still significant [64], and is considered complementary when the two variables point in the same direction [51]. In this case, the inclusion of the variable Organisational Commitment/Work Engagement between the variables Occupational Health and Safety and Turnover Intention reduced the strength of the direct relationship between these two variables from −0.443 to −0.207 (Figure 1 and Figure 2) while maintaining the same sign.

To calculate the magnitude of an indirect effect, one must consider the value of the variance accounted for (VAF) [57], which determines the size of the total indirect effect. In this case, we calculated a VAF of 0.529, meaning partial mediation and a magnitude of 0.529.

To evaluate and validate the mediation model, we used the bootstrapping method [67]. Bootstrapping is a non-parametric procedure for mediation analysis, for both a single variable and multiple variables, that is valid for small sample sizes and is therefore suitable for the PLS-SEM method [57]. Among the aspects evaluated for both direct and indirect effects were the magnitude of these effects, their confidence interval, the t-value, and significance based on the *p*-value [51]. These values are shown in Table 3.

The data analysed prove the existence of a significant complementary partial mediation between these variables.

## 4. Discussion

According to the results, health and safety at work has a direct positive influence on organisational and work engagement (Hypothesis 1) and a negative influence on the intention to leave the company (Hypothesis 3). Organisational and work engagement was also found to have a direct negative impact on employees’ intention to leave the company (Hypothesis 4). It was also found that organisational and work commitment significantly mediates between health and safety at work and employees’ turnover intention (Hypothesis 2).

These results show that, even in occupational sectors seen as non-hazardous, occupational health and safety policies are quite important to employees’ turnover intention, a result that is in line with that observed in previous studies in different occupational sectors [20,26,42,68].

This is, in short, a quite logical reaction: workers who feel that their employer is concerned about their health and safety will be more reluctant to leave the organisation, even in low-risk sectors such as IT consultancy [20,26,42,68].

These data serve to confirm the first hypothesis of our work, namely the existence of a relationship between occupational health and safety policies and employees’ turnover intention.

It was also observed that occupational health and safety policies have a positive impact on employees’ organisational and work-related commitment, which is also in accordance with previous work [21,42].

These data confirm our second hypothesis, namely the existence of a relationship between occupational health and safety policies and employees’ work and organisational commitment.

As in the case of the previous hypothesis, this is also to be expected: if workers feel that their employer cares about their health and safety, they will be more committed to both their work and their organisation, even in a low-risk sector as in this case.

We also observed the existence of a negative relationship between organisational and work commitment and intention to leave the company, which implies that employees who are highly committed to their work and their company are reluctant to leave the company; this is in agreement with the findings of previous studies [44,45].

This supports our third hypothesis, namely the existence of a relationship between employees’ work and organisational commitment and their intention to leave the company.

While the latter was also expected, the inclusion of the mediating effect indicates that employees who are satisfied with their company’s health and safety policies are more likely to be committed to the company and, consequently, to have lower levels of turnover intention. This demonstrates that work and organisational commitment mediates between health and safety policies and employee turnover.

## 5. Conclusions

According to the data analysed in this study, it can be affirmed that employees who are satisfied with the occupational health and safety policies at work tend to have a higher degree of commitment to both their organisation and their work, which decreases their intention to leave the company. On the contrary, employees who are not satisfied with these policies tend to have a lower degree of commitment to the organisation and the job, thus increasing their interest in leaving the company.

It is also clear that this is the case even in low-risk sectors, such as IT consultancy in Spain, for which occupational health and safety has not usually been considered a priority. As we have seen previously, this sector also has its risks, which are mainly of a psychological nature.

On the basis of this study, and considering both the high level of turnover in the sector and its strategic importance both from an economic point of view and in terms of employment creation, there is a need, both on the part of the management of these companies and on the part of different public administrators, to strengthen and encourage occupational health and safety policies in the sector.

While it is clear that health and safety policies improve the organisational commitment of workers and decrease turnover rates, we consider it important not only to implement and improve these policies, but also to work on making them known to workers.

It is true that previous studies have analysed the impact of job security on organisational commitment and turnover intention, but no studies were found that analyse the impact of these factors on the national IT consultancy sector. Therefore, this study may fill this gap and help organisations implement appropriate policies to minimise the impact of the high turnover rates in the sector.

Although this study has been able to test the selected working hypotheses, it is limited only to the impact of occupational health and safety policies at work. It is important to point out that, in addition, this study was carried out in the middle of the crisis caused by the Covid-19 pandemic. This may have had an impact on workers’ perception of these policies, as this sector has traditionally not given great importance to safety, and this situation changed during the pandemic period. Another possible limitation of the study is that we did not obtain a high number of complete responses, which conditions the scope of the analysis.

From a practical point of view, what this study teaches us is that investment in health and safety helps organisations to improve organisational commitment and reduce employee turnover, especially in a situation such as that caused by Covid-19 in which health becomes even more important. In our opinion, actions such as facilitating the possibility of teleworking in circumstances such as the current ones may increase the organisational commitment of employees, as they maintain their health, which has been the number-one objective of people during the Covid-19 pandemic. Not having access to company data collected before and after implementing an occupational health and safety plan limits our work by not allowing us to determine how organisational commitment and turnover intention evolve over time.

Future studies will investigate the impact that other factors such as organisational support, independence at work, job satisfaction, stress, and emotional independence have on the well-being of workers and their intention to leave the company [69,70,71,72], thus achieving a broader analysis that includes different work-related factors and helping us to understand the relationships between them more broadly.

## Figures and Tables

**Figure 1 ijerph-18-05658-f001:**
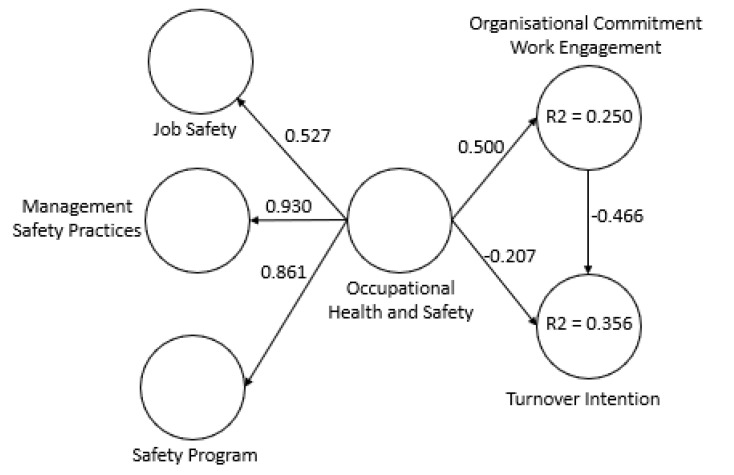
Structural equation model.

**Figure 2 ijerph-18-05658-f002:**
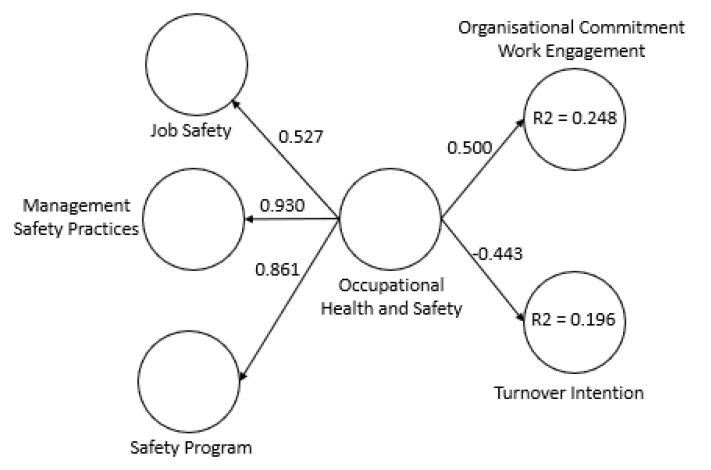
Structural equation model without mediation.

**Table 1 ijerph-18-05658-t001:** Factor loadings, means, standard deviations, reliabilities, and average variance extracted.

Construct	Item	Mean	SD	Factor Loading	Cronbach’s Alpha	CR	AVE
Job Safety	R1(SQ001)	3.188	1.312	0.856	0.849	0.899	0.690
	R1(SQ002)	3.673	1.176	0.766			
	R1(SQ003)	2.939	1.323	0.895			
	R1(SQ004)	2.377	1.186	0.800			
Management Safety Practices	W1(SQ001)	3.372	1.206	0.860	0.952	0.960	0.726
	W1(SQ002)	3.250	1.278	0.812			
	W1(SQ003)	3.573	1.136	0.857			
	W1(SQ004)	3.400	1.222	0.773			
	W1(SQ005)	3.613	1.162	0.903			
	W1(SQ006)	3.603	1.160	0.876			
	W1(SQ007)	3.786	1.144	0.797			
	W1(SQ008)	3.802	1.116	0.886			
	W1(SQ009)	3.661	1.181	0.894			
Safety Programme (Policies)	X1(SQ001)	3.804	1.024	0.849	0.948	0.957	0.762
	X1(SQ002)	3.880	0.933	0.901			
	X1(SQ003)	3.791	1.035	0.938			
	X1(SQ004)	3.803	0.953	0.909			
	X1(SQ005)	3.431	1.055	0.833			
	X1(SQ006)	3.822	1.036	0.838			
	X1(SQ007)	3.707	1.015	0.837			
Organisational Commitment	I1(SQ001)	3.487	1.024	0.802	0.842	0.886	0.611
	I1(SQ002)	3.854	0.976	0.847			
	I1(SQ003)	3.960	1.004	0.765			
	I1(SQ004)	3.247	1.331	0.826			
	I1(SQ005)	3.359	1.187	0.653			
Turnover Intention	N10(SQ001)	2.919	1.341	0.928	0.765	0.862	0.683
	N10(SQ002)	3.756	1.107	0.594			
	N10(SQ003)	3.359	1.384	0.914			

**Table 2 ijerph-18-05658-t002:** Discriminant validity—Fornell–Larcker criterion.

	Job Safety	Management Safety	Safety Programme	Organisational Commitment	Turnover Intention
Job Safety	0.831				
Management Safety	0.393	0.852			
Safety Programme	0.296	0.656	0.872		
Organ. Commitment	0.369	0.488	0.359	0.782	
Turnover Intention	−0.569	−0.428	−0.289	−0.569	0.826

**Table 3 ijerph-18-05658-t003:** Direct and indirect effects.

Direct Effect	95% Confidence Interval for Direct Effect	*t*-Value	Significance (*p* < 0.05)
−0.207	(−0.335, −0.057)	2.944	Yes
Indirect effect	95% confidence interval for direct effect	*t*-value	Significance (*p* < 0.05)
−0.233	(0–299, −0.171)	7.011	Yes

## Data Availability

Not applicable.

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
