# Peer review of "Occupational Health and Safety, Organisational Commitment, and Turnover Intention in the Spanish IT Consultancy Sector"

_ijerph, 2021, doi:10.3390/ijerph18115658_

Round 1
Reviewer 1 Report
The authors should address the significance of the study and further define the research gap and methodology.
Author Response
Good afternoon, I attach a document with the answer.Best regards

Reviewer 2 Report
Many thanks for the effort made at presentine the work. To help improve the paper quality consider the follwoing:
i. All through the paper consider further proof read to help improve the presentation
ii. I presume the use of the word "affective" was mistaken for "effective" if so consider revising this all through the paper.
iii. Line 181-194. The content here should be moved to the result section considering the content is providing insight to the partcipant demograhic charactersitics and not method used in colecting the data from the partcipants.
iv. Line 267-272. The statement at the start need looking at to help improve it intent.
v. Line 306-313 replace the phrase paragraph to help improve it content. The use of word such as "we can see" should be replaced all through in my opinion.
vi. There has been frequent use of the word "intention to change" i.e. lines, 76, 82, 315 etc. It this change along the line of staff tunover or adapt to safety culture within the orgabisation. This need clarifying further.
vii. Line 327-329 " Estos datos ratifican nuestra segunda hipótesis, la existencia de una relación entre las políticas de seguridad y salud laboral y el compromiso con el trabajo y la empresa de los empleados". I there is no direct link with the rest of the paper.
Author Response
Good afternoon,
I attach a document with the answer.
Best regards

Round 2
Reviewer 1 Report
Nil
Author Response
I am attaching a response to corrections.Kind regards

Reviewer 2 Report
Many thanks for the update provided. I still find the paper needing further improvement.
i. The instroduction section has now become unnecessary long and in my opinion it stand the risk of loosing readers interest. Advised this be trimmed down.
ii. Section 2 Materials and Methods. The section added (line 206-226) in my opinion did not provide direction on how PLS-SEM was applied in the study but rather further justification for the approach adoption was rather provided. This need updating .
iii. As stated in my previous comment , the use of first person should be avioded were possible. However this has been actively utilised all through the paper i.e. "we" . To help improve the paper quality, aduthors are advised to tone this down.
iv. Line 417-420. It was not clear the intent of this statement . Advise this need revising.
v. Overall, i will recommend the paper be further prood read to help improve the grammar structure.
I will be happy to go through the paper once these corrections are considered.
Author Response
Good afternoon, I am attaching a response to corrections.Kind regards
